# THz Filters Made by Laser Ablation of Stainless Steel and Kapton Film

**DOI:** 10.3390/mi13081170

**Published:** 2022-07-25

**Authors:** Molong Han, Daniel Smith, Soon Hock Ng, Zoltan Vilagosh, Vijayakumar Anand, Tomas Katkus, Ignas Reklaitis, Haoran Mu, Meguya Ryu, Junko Morikawa, Jitraporn Vongsvivut, Dominique Appadoo, Saulius Juodkazis

**Affiliations:** 1Optical Sciences Centre and ARC Training Centre in Surface Engineering for Advanced Materials (SEAM), School of Science, Swinburne University of Technology, Hawthorn, VIC 3122, Australia; molonghan@swin.edu.au (M.H.); danielsmith@swin.edu.au (D.S.); tkatkus@swin.edu.au (T.K.); haoranmu@swin.edu.au (H.M.); saulius.juodkazis@gmail.com (S.J.); 2Melbourne Centre for Nanofabrication, 151 Wellington Road, Clayton, VIC 3168, Australia; 3Australian Centre for Electromagnetic Bioeffects Research, Swinburne University of Technology, Hawthorn, VIC 3122, Australia; zvilagosh@swin.edu.au; 4Institute of Physics, University of Tartu, 50411 Tartu, Estonia; vijayakumar.anand@ut.ee; 5Institute of Photonics and Nanotechnology, Vilnius University, Saulėtekio Ave. 3, 10257 Vilnius, Lithuania; ignas.reklaitis@gmail.com; 6National Metrology Institute of Japan (NMIJ), National Institute of Advanced Industrial Science and Technology (AIST), Tsukuba Central 3, 1-1-1 Umezono, Tsukuba 305-8563, Japan; ryu.meguya@aist.go.jp; 7WRH Program International Research Frontiers Initiative (IRFI) Tokyo Institute of Technology, Nagatsuta-cho, Midori-ku, Yokohama 226-8503, Japan; morikawa.j.aa@m.titech.ac.jp; 8CREST-JST and School of Materials and Chemical Technology, Tokyo Institute of Technology, Ookayama, Meguro-ku, Tokyo 152-8550, Japan; 9ANSTO—Australian Synchrotron, Infrared Microspectroscopy (IRM) Beamline, 800 Blackburn Road, Clayton, VIC 3168, Australia; jitrapov@ansto.gov.au; 10ANSTO—Australian Synchrotron, THz/Far-IR Beamline, 800 Blackburn Road, Clayton, VIC 3168, Australia; dominiqa@ansto.gov.au

**Keywords:** THz filters, synchrotron infrared, anisotropy

## Abstract

THz band-pass filters were fabricated by femtosecond-laser ablation of 25-μm-thick micro-foils of stainless steel and Kapton film, which were subsequently metal coated with a ∼70 nm film, closely matching the skin depth at the used THz spectral window. Their spectral performance was tested in transmission and reflection modes at the Australian Synchrotron’s THz beamline. A 25-μm-thick Kapton film performed as a Fabry–Pérot etalon with a free spectral range (FSR) of 119 cm${}^{-1}$, high finesse $F_{c}\approx 17$, and was tuneable over ∼10μm (at ∼5 THz band) with $\beta =30^{\circ }$ tilt. The structure of the THz beam focal region as extracted by the first mirror (slit) showed a complex dependence of polarisation, wavelength and position across the beam. This is important for polarisation-sensitive measurements (in both transmission and reflection) and requires normalisation at each orientation of linear polarisation.

## 1. Introduction

The spectral range available on the THz/Far-IR beamline is extremely broad (1 μm to 1 mm in wavelength), and as such, it requires different types of spectral filters and optical elements to manipulate its polarisation behaviours, involving spin angular momentum (SAM) and orbital angular momentum (OAM), of this broadband radiation. In some bio-medical applications, it is imperative to be able to distinguish a certain part of the THz-IR spectral range that is effective for specific analysis. This is particularly important for the beam that carries the SAM and OAM properties since their absorption is dependent on microscopic structures as well as the chirality of constituent molecules/compounds. It is also useful to carry out proof-of-concept prototyping experiments prior to the design and fabrication of required optical elements (e.g., filter, polariser, and waveplate). Therefore, the key motivation of this study was to investigate and to gain a better understanding of the beam profiles as a result of these high-transmission THz band-pass filters [1,2,3,4], which were produced by a simple laser ablation fabrication method. A short UV wavelength ps-laser radiation was previously used to cut out mesh filters from a metallic foil, as well as to ablate metal coating from a polymer substrate [5]. When dielectric of the refractive index *n* fills the opening aperture of the crosses after UV-laser ablation of the metal coating, the central frequency of the filter is red-shifted by a factor of $\sqrt{\frac{2}{n^{2}+1}}$ [5]. The same method was used to ablate cross-patterns combined into the Fresnel zone plate for THz focusing at specific wavelengths [6], which was also demonstrated by patterning of graphite on flexible substrates [7]. Using sub-0.5 ps pulses at ∼1μm wavelength focused by a telecentric lens under galvanometric scanning, a superior shape control of the side wall angle (with the surface normal) can be achieved [8]. This is important for high-volume manufacturing, as well as for gaining control of the surface the roughness, which reduces conductivity $\sigma_{ef}$ due to roughness factor $K_{SR}$: $\sigma_{ef}=\sigma_{0}/K_{SR}^{2}$, where $\sigma_{0}$ is for smooth surface, affecting filter performance at lower frequency ∼0.1 THz window [9]. Polarisation-sensitive converters based on L-shaped and split-ring apertures were also demonstrated by ns-laser ablation of the foils [10].

The performance of our optical elements was assessed on the THz/Far-IR beamline at the Australian Synchrotron (AuSy) after the first mirror used for extracting IR radiation from the storage ring was replaced in January 2022, and both optics and beam propagation were subsequently re-aligned to achieve optimal throughputs in May 2022 (Figure 1a,b). A detailed analysis of wavelength and polarisation behaviors over the cross section of incident synchrotron radiation will play a critical role for emerging applications in the field of 3D polarisation tomography using near-field attenuated total reflection (ATR) apparatus, which has recently been demonstrated for characterising biological samples in the IR to THz spectral range [11]. Polarisation and wavelength composition of the synchrotron IR/THz beam at the focal point has a complex structure, mainly because the synchrotron is not a point source, but has a longitudinal dimension. This unique beam shape possesses complex optical properties due to magnetic field contributions of the edge radiation (ER) and bending magnet radiation (BMR), resulting from different locations of the bending trajectories of electrons [12] (see Supplement for additional information based on the 1st mirror used since the first synchrotron-IR light in 2007). According to our previous investigation in 2018, the synchrotron beam at the THz/Far-IR beamline has a combination of both linear (22%) and circular (78%) polarisations due to the contribution of the BMR and ER radiations, respectively [12].

Both absorption coefficients and refractive indices of water and biological tissues are dominated by water, undergoing significant changes in the wavenumbers $\tilde{\nu }=20$ to 600 cm${}^{-1}$ (0.6 to 18 THz) region. The absorption coefficient α=4πκ/λ, where κ is the imaginary part of the refractive index $\tilde{n}=n+i\kappa$ (permittivity ε is $\tilde{n}\equiv \sqrt{\varepsilon }$). The optical density OD is defined as $e^{-\alpha d}=10^{-OD}$ for a sample of thickness *d* (OD=αd/ln10). The refractive index reduces from n∼2.19 at $\tilde{\nu }=20$ cm${}^{-1}$, to n∼1.33 at 600 cm${}^{-1}$, with the absorption coefficient increasing from α∼175 cm${}^{-1}$ at $\tilde{\nu }=20$ cm${}^{-1}$ to α∼3200 cm${}^{-1}$ at 600 cm${}^{-1}$ at room temperature. The rapid change in water properties presents a challenge as varying frequencies have vastly different tissue penetration properties defined by the skin depth $\delta_{s}=1/\alpha$. This complicates the assessment of thermal and non-thermal effects of radiation on tissues, as well as changes of spectral properties of biological samples upon water freezing [13,14,15,16]. In this aspect, the availability of narrow band filters will assist for effective characterisation of biological tissue properties that are needed for diagnostic applications.

Here, we demonstrate a simple method for fabricating such THz filters from Kapton (polyimide) film and stainless steel micro-thin foils. Laser cut by ablation was carried out at a high intensity of ∼1 PW/cm${}^{2}$/pulse using femtosecond (fs) laser machining [17,18]. In this study, the fs-laser ablation was performed without change of focal position along the beam. This was possible since the depth-of-focus was comparable to the thickness of the film used to cut the cross filters. This process also made all the laser fabrication simpler and faster. The performance of these filters was subsequently assessed on the THz/Far-IR beamline at the AuSy in both transmission and reflection setups.

## 2. Samples and Methods

### 2.1. Laser Cutting of Filters

The cross-shaped apertures were fabricated by fs-laser ablation cutting. This consisted of a 10 W average power PHAROS laser (Light Conversion Ltd. Vilnius, Lithuania) coupled with a three-axis positioning stage controlled by SCA software and integrated with a 3D laser machining station (Workshop of Photonics, Ltd. Vilnius, Lithuania). Two materials were used to fabricate the filters, including 25-μm-thick SUS304 stainless steel foil (Jianglin Steel Corporation PTE Ltd. Tianjin, China) and 25-μm-thick Kapton (polyimide derived from pyromellitic dianhydride and 4,4-oxydianiline; Du Pont-Toray, Co., Ltd. Tokyo, Japan [19]). The laser beam was scanned multiple times along the contour line of the cross until each was completely cut through. The main parameters of the laser system include the laser wavelength λ=1030 nm, pulse duration $t_{p}=230$ fs, pulse energy $E_{p}=50\;\mu$J/pulse laser (on the sample), and repetition rate $f_{p}=200$ kHz. Additionally, the scanning speed was set at $v_{s}=50$ mm/s. A vacuum suction nozzle was positioned close to the cutting area to remove ablated particles and the cutout cross pieces. The process required 6 passes to completely cut through the SUS304 foil, while only 3 passes were required for the Kapton film at half of the pulse energy (i.e., 25μJ/pulse). The focal position was placed on the surface of the sample and remained the same for subsequent passes.

The numerical aperture of the objective lens (Mitutoyo) was NA=0.26, which focused the laser beam to a 2r=1.22λ/NA=4.8μm focal spot. Depth of focus can be estimated as double the Rayleigh length $2z_{R}=2\frac{\pi r^{2}}{\lambda }=35.6\;\mu$m, which is larger than the thickness of the samples. The pulse fluence was $F_{p}=E_{p}/\left(\pi r^{2}\right)=272.5$ J/cm${}^{2}$ (for pulse of 50 μJ), which is more than $10^{3}$ times the laser ablation threshold of metals at ∼0.1 J/cm${}^{2}$. The pulse average intensity $I_{p}=F_{p}/t_{p}=1.19$ PW/cm${}^{2}$ (ablation threshold of metal ∼0.4 TW/cm${}^{2}$). This is a high irradiance and hard X-ray generation takes place in the plasma region of the target due to bremsstrahlung [20]. When targets with heavy elements, such as metal targets, are used, hard X-ray generation can be significant [21,22]. It was shown that the personal exposure dose rate was $\dot{H}\left(0.07\right)\approx \left(1-2\right)$ [mSv/h/W] at 20 cm distance from plasma, which saturates for the stainless steel targets at an irradiance of 0.1 PW/cm${}^{2}$ with different pulse duration but using comparable exposure conditions as in our study [23]; the depth of exposure $d_{p}=0.07$ mm in $\dot{H}\left(d_{p}\right)$. According to the recommendation of the International Commission of Radiological Protection (ICRP), the effective dose limit in planned exposure situations is 20 mSv per year for occupational exposures averaged over a period of five years and only 1 mSv/year for a visitor [24]. At an $f_{p}=200$ kHz repetition rate and scanning speed of $v_{s}=50$ mm/s, the pulse-to-pulse distance is $d_{pp}=v_{s}/f_{p}=250$ nm or 5.2% of the focal diameter 2r=4.8μm. This corresponds to a strong overlap between adjacent pulses.

Due to a large area fabrication 5×5 mm${}^{2}$, plane fitting helps to keep the focal spot on the surface of the sample during long laser cutting times. The procedure described below can be used for initial plane alignment to dynamically change height (z-position) or repeated during long fabrication. In this study, due to a comparatively large depth-of-focus (two Rayleigh lengths), the plane tilt was not critical and was only aligned at the outset of fabrication. First, coordinates of focal spot $P_{i}\left(x_{i},y_{i},z_{i}\right)$ on the surface of sample are determined at three corners of the write field; i=1…3. Two vectors $\vec{P_{1}P_{2}}=\left(x_{2}-x_{1},y_{2}-y_{1},z_{2}-z_{1}\right)$ and $\vec{P_{1}P_{3}}=\left(x_{3}-x_{1},y_{3}-y_{1},z_{3}-z_{1}\right)$ are in the plane and the normal to the plane is:(1)$$\vec{P_{1}P_{2}}\times \vec{P_{1}P_{2}}=\left|\begin{array}{ccc} \vec{i} & \vec{j} & \vec{k}\\ x_{2}-x_{1} & y_{2}-y_{1} & z_{2}-z_{1}\\ x_{3}-x_{1} & y_{3}-y_{1} & z_{3}-z_{1} \end{array}\right|=\langle a_{n},b_{n},c_{n},\rangle .$$

The plane through the point $P_{1}$ (also $P_{2}$ and $P_{3}$) is $a_{n}\left(x-x_{1}\right)+b_{n}\left(y-y_{1}\right)+c_{n}\left(z-z_{1}\right)=0$. This is the plane equation which will account for the actual tilt. During laser writing, for the each new point $P_{new}\left(x_{new},y_{new}\right)$ along writing trajectory, the height $z_{new}$ is calculated from the plane equation given above. The plane fitting method keeps the focal spot on the surface of the sample when there is a uniform tilt. Such tilt compensation methods are implemented in direct write lithographies, e.g., electron beam lithography. For large area fabrication, the sample-focus position can drift over time. It is possible to repeat the described procedure when required manually or automatically when the intensity of back-reflected light is monitored. In the future, surface mesh leveling can be used to maintain the focal spot over a larger area by taking into account the surface topology.

In case of the filters made out of Kapton, the cross mesh pattern was coated with 20 nm-thick chromium and 50 nm-thick gold. This thickness is close to the skin depth, which reduces intensity of THz radiation by 1/e-times (detailed discussion in Section 3.2).

### 2.2. Near-Normal Incidence Transmission and Reflection Measurements

In order to determine optical properties of THz filters, measurements of reflected and transmitted intensities, $I_{R}$ and $I_{T}$, respectively, were made from exactly the same position on each filter. This was possible with the use of a Near-Normal Incidence Transmission and Reflection Optics (N${}^{2}$ITRO, Bruker) unit (Figure 2). The diameter of the THz beam on the sample was ∼2 mm (Figure A3). Polarisation over the area of the focal spot was non-uniform as expected from the synchrotron radiation; see Supplement for definition of Stokes vector for an electron of a spinning trajectory around the magnetic field.

The measurements of transmittance $T=\frac{I_{T}}{I_{0}}$ and reflectance $R=\frac{I_{R}}{I_{0}}$ were measured from the same spot on the filter at a selected polarisation angle θ; $I_{0}$ is the incident THz intensity. Following energy conservation, the absorptance A=1−R−T. We determined the optical density OD of the composite films with the following expression $T=\left(1-R\right)\times 10^{-OD}$, where *R* and *T* were directly measured; also $A=\left(1-R\right)\times \left(1-10^{-OD}\right)$. Strong absorption conditions, when αd>1, is defined by the absorption coefficient α [cm${}^{-1}$] and thickness *d* as αd=ln(10)OD≡2.303×OD.

The lesser explored spectral region at 0.1–1 mm (100–10 cm${}^{-1}$, 3–0.3 THz) wavelengths was selected for measurements and band-pass filters (usually, a Globar source is used in table-top FTIR spectrometers for measurements at wavenumbers larger than $\tilde{\nu }>400$ cm${}^{-1}$). Due to strong interference in the 120 μm Mylar beam splitter, the low intensity regions have higher $T(\tilde{\nu })$ and $R(\tilde{\nu })$ spectra (Figure A1).

The reflectance $R=\left[{\left(n-1\right)}^{2}+\kappa^{2}\right]/\left[{\left(n+1\right)}^{2}+\kappa^{2}\right]$ at normal incidence to an air/vacuum interface and the complex refractive index of the sample is $\tilde{n}=n+i\kappa$. Lower reflectance *R* contributes to higher absorptance $A=\left(1-R\right)\times \left(1-10^{-OD}\right)$.

With the portions of reflected *R* and transmitted $T=\left(1-R\right)10^{-OD}$ light (energy), the absorbed portion is determined $A=\left(1-R\right)\left(1-10^{-OD}\right)$; the energy conservation A+R+T=1 holds. Hence, the absorption coefficient α=4πκ/λ (for intensity) can be determined, with both the complex refractive index and permittivity (dielectric susceptibility ε) related with $\sqrt{\varepsilon }=n+i\kappa$. With *n* and κ determined, the response of material at the chosen wavelength is known. For bulk material (not film) with no transmission, i.e., T=0, $A=1-R=2\kappa^{2}/\left[{\left(n+1\right)}^{2}+\kappa^{2}\right]$ since $R=\left[{\left(n-1\right)}^{2}+\kappa^{2}\right]/\left[{\left(n+1\right)}^{2}+\kappa^{2}\right]$.

## 3. Results and Discussion

### 3.1. Cross-Filters Out of Metal Foil

Band-pass filters are essential for determining the absorption effect at specific wavelengths. The scaling for the design of such binary transmission filters, where the central wavelength is defined by period *P*, width of opening *W*, and length of cross opening *L* as $\lambda_{c}=1.8L-1.35W+0.2P$, is established [2]. For filters aimed at a low wavenumber spectral range (0.3–3 THz), all dimensions (*W*, *L*, and *P*) are within 10–100 μm. Direct laser writing based on ultra-short laser pulses was used to cut the openings in micro-films of stainless steel SUS304 and Kapton used in this study (Figure 1c).

Figure 3 shows the spectral performance (a) and detailed structure (b) of the filters cut from SUS304 foil. Both, 1 and 2 THz filters as designed, showed transmission localised at slightly lower frequencies of 0.85 and 1.66 THz, respectively. The most sensitive term defining $\lambda_{c}=c/\nu_{c}$ is 1.8 L. The finite width of the laser cut increases the length of the opening, causing larger $\lambda_{c}$ (smaller frequency $\nu_{c}$). The cross-shape filters were expected to be independent on the polarisation of incident THz beams; however, a clear dependence of transmission on the polarisation was observed. Each transmission spectrum was normalised to the spectrum without sample, while reflectance spectra were normalised to reflection of an Au-mirror. It is important to note that each peak in the transmission spectra coincided with a dip in reflectance as expected (not plotted to avoid clutter). Both *T* and *R* show significant dependence on the beam polarisation; this suggests that each individual spectra should be measured as references to decouple material-dependent and radiation-dependent effects. The polarisation distribution over the focal spot is not uniform as was revealed in a previous inspection (see Appendix A). The isotropic edge component and linear dipole parts of radiation are present and reflected from a two-lobed mirror, which fills the focal volume with different spectral components directed along slightly different wavevectors and having isotropic and linear polarisations.

### 3.2. Cross-Filters and Fabry-Pérot Etalon Out of Kapton Film

Simple and fast laser cutting of the required cross-shapes was made out of a 25-μm-thick Kapton film. The film was coated with 20 nm of Cr and 50 nm of Au to make a binary transmission mask. A very similar transmission to SUS304 filters was confirmed (Figure 4). This shows that there were no geometrical differences in the definition of the cut width and edge quality between very different materials. The complexity of polarisation and wavelength distribution might be contributing to lower transmittance due to strong diffraction (see Supplement for the Stokes parameters of emitted radiation from an electron spinning around magnetic field). The ratio of open area per unit cell of the pattern is $R_{A}=Open/Cell=\left(2LW-W^{2}\right)/P^{2}=23\%$ for the ∼2 THz filter design.

The skin depth for EM radiation in an absorbing conductive coating can be estimated from $\delta_{s}\equiv 1/\alpha =\sqrt{\frac{2\rho }{\omega \mu }}$ when $\nu \ll \frac{1}{\rho \epsilon }$ and $\delta_{s}\approx 2\rho \sqrt{\frac{\epsilon }{\mu }}$ when $\nu \gg \frac{1}{\rho \epsilon }$, where $\epsilon =\epsilon_{0}\epsilon_{r}$ is the permittivity of the coating, $\mu =\mu_{0}\mu_{r}$ is its permeability, ρ [Ωm] is the resistivity, ω=2πν is the cyclic frequency of light, $\epsilon_{0},\mu_{0}$ are the permittivity and permeability of free space, respectively. For Cr $\rho_{Cr}=13\times 10^{-8}$ [Ωm] or Au $\rho_{Au}=2.24\times 10^{-8}$ [Ωm] with $\epsilon_{0}=\frac{1}{\mu_{0}c^{2}}=8.85\times 10^{-12}$ [F/m] and $\epsilon_{r}={\tilde{n}}^{2}$, the ν=(1−10) THz frequencies are always smaller than 1/(ρϵ). Hence, the skin depth at which intensity of light is reduced $1/e^{2}=7.4$-times is defined by $\delta_{s}=\sqrt{\frac{2\rho }{\omega \mu }}=\frac{503}{\sqrt{\mu_{r}\nu \sigma }}$ where conductivity σ=1/ρ [S/m] and $\mu_{r}=1$ for non magnetic materials, e.g., gold. For gold at ν=1 THz, one would find $\delta_{s}=75$ nm and for 10 THz $\delta_{s}=24$ nm. Chromium is antiferromagnetic below 38 ${}^{\circ }$C and for an estimate we consider $|\mu_{r}|∼1$, which yields in $\delta_{s}\approx 181$ nm (1 THz) and 57 nm (10 THz). We used Cr/Au:20 nm/50 nm coating on Kapton, which is close to the skin depth at several THz frequencies. The THz filter made out of Kapton coated by a cummulative metal layer of 70 nm (Cr and Au) had transmittance slightly lower as compared with the stainless steel filter. From the presented skin depth analysis, one could expect a better contrast of transmittance at the calculated band for a thicker (opaque) metal film. However, it is not imperative to use tens-of-micrometers metal foil for high transmittance and contrast filters. Due to faster fabrication of filters out of Kapton, coating is an appealing simplification in THz filter fabrication.

When a non-coated Kapton filter (original film) was inserted into the beam (sample in N${}^{2}$ITRO unit), clear Fabry–Pérot (FP) etalon action was observed (Figure 4) with a free spectral range of FSR=119 cm${}^{-1}$ as defined by d=25μm thickness of the Kapton film. The ratio of FSR to the bandwidth Δλ=7 cm${}^{-1}$ defines the finesse $F_{c}$, which is also the number of wavelengths which can be resolved by FP etalon $F_{c}=FSR/\Delta \lambda_{c}=17$ at this THz spectral range. From the known thickness of Kapton d=25μm and $FSR=\frac{1}{2nd}$ cm${}^{-1}$ = $\frac{\lambda^{2}}{2nd}$ nm = $\frac{c}{2nd}$ Hz, one finds the refractive index of Kapton n=1.68. By changing the thickness of the Kapton film, the bandwidth Δλ∝1/d can be changed. Spectral positioning of the FP resonances can be tuned by tilt angle β of the FP etalon according to $2d\cos\beta =m\lambda_{c}$, where the *m* is order parameter (an integer); see inset in Figure 4a,b. It can be efficient to use FP etalon as a wavelength selection filter, especially for selective energy delivery to a sample by absorption. Indeed, Δλ=7 cm${}^{-1}$ at $\lambda_{c}=350$ cm${}^{-1}$ comprises only 2% bandwidth, which is smaller by approximately one order of magnitude as compared to cross-filters. Since higher orders of FP etalon converge towards the centre of the detector, it is a welcoming feature since it is less sensitive to beam alignment. The interference which caused the FP maxima was 1.47 times larger than the normalised transmission T=1 due to the constructive addition of FP modes. It is noteworthy that R→0 at the T→max (Figure 4a), a tendency recognisable in the experiment.

### 3.3. Concept of Tunable THz Filters

Figure 5 shows the concept of using the N${}^{2}$ITRO unit as a tunable filter when the FP etalon is inserted into the sample’s position at a controlled tilt angle β. The effective thickness of the FP etalon (d=25μm Kapton in this study) becomes larger with tilt d/cosβ, which tunes the phase delay δ=2πnd/λ. The transmission and reflection coefficients of the FP etalon are dependent on the reflectivity *R* of the FP film interfaces (both interfaces are assumed to have the same reflectivity) and FP transmittance and reflectance spectra are given by $T=\frac{{\left(1-R\right)}^{2}}{{\left(1-R\right)}^{2}+4R\sin^{2}\delta }$, $R=\frac{4R\sin^{2}\delta }{{\left(1-R\right)}^{2}+4R\sin^{2}\delta }$, respectively (Figure 5). The FP etalon imparts its absorbance spectrum onto the synchrotron radiation, but using the N${}^{2}$ITRO geometry allows the same ∼2 mm diameter point to be probed in *R* and *T*. A sample’s absorbance can be measured under broad-band *R* and narrow-band *T* excitation. Since polymer films of a few micrometer thickness are readily available (e.g., 4-μm-thick ultralene), the FSR∝1/d can be considerably increased from the 119 cm${}^{-1}$ which was for d=25μm Kapton film.

### 3.4. Polarisation Analysis of Reflectance Spectra

The N${}^{2}$ITRO unit is useful for measurements of optically thick samples (T→0) in reflection. Such a measurement is more sensitive to the real part of refractive index *n* since $R=\left[{\left(n-1\right)}^{2}+\kappa^{2}\right]/\left[{\left(n+1\right)}^{2}+\kappa^{2}\right]$ at normal incidence to an air/vacuum interface and $\tilde{n}=n+i\kappa$ being the complex refractive index of the sample [25]. This is compared to absorbance $A=1-R=2\kappa^{2}/\left[{\left(n+1\right)}^{2}+\kappa^{2}\right]$ when T=0. To test this spectral property, samples were prepared from a thermally fused bundle of optical silica fibers, which are available as chatoyant “cat-eye” souvenir silica stones (Figure 6a); color pigments are added to the bundle before melting to stain the stones; however, the central silica core remains transparent and colorless. The as-cut and as-polished samples were mounted into the N${}^{2}$ITRO holder with vertical alignment (along the y-axis; perpendicular to the 1st-mirror slit along the x-axis; Figure 2).

Strong polarisation dependence of reflectance *R* was observed with clear dispersion-like spectral lineshape around the Si-O-Si symmetric bending band at 480 cm${}^{-1}$ [26]. Interestingly, this is the exact spectral position of the characteristic feature in Raman scattering spectra of silica glasses, which is understandable since scattering is sensitive to the refractive index. Since measurements were carried in reflection, which is more sensitive to the refractive index, the form birefringence of optical fibers dominates the spectral lineshape. Reflectance changed ∼5% for spectra along and across silica fibers at the most sensitive part of the spectrum near the absorption band; however, anisotropy of *R* was present over the entire THz spectral window. Birefringence as an anisotropy in refractive index is revealed in reflectance spectra of the sample with aligned fiber pattern. When direct transmission cannot be measured (a thick sample) and when there is no dichroism in the sample, it is still possible to reveal anisotropy in material distribution (fibers in this study) since by definition the refractive index is proportional to the mass density. It was critical to normalise each *R* spectra to the background measured with an Au mirror at the same input polarisation, since polarisation and wavelength distribution over the illumination area is complex (Figure A3). For the glass the refractive index n≈2.5 at THz spectral window (see discussion of Figure A4b in Appendix A), reflection coefficient is $R=\frac{{\left(n-1\right)}^{2}}{{\left(n+1\right)}^{2}}\approx 18.4\%$ and an increase of *n* by Δn=0.25 would cause R=21.8%, close to the experimental observation (Figure 6). It is shown here that form-birefringence can be recognised at the THz spectral range from subtle changes in Reflectance over a spectrally broad spectral region where absorption bands are absent. Moreover, dispersion-like spectral lineshapes identifies related absorption bands and can be revealed by polarisation-sensitive *R* measurements (Figure 6).

## 4. Conclusions and Outlook

THz filters with a bandwidth of $\Delta \lambda /\lambda_{c}∼10\%$ can be made by laser cutting patterns of crosses in metals or polymer foils. Coating ∼100 nm of metal onto polymers corresponds to skin depth and renders them optically opaque. Laser cutting of micro-films can be made with high throughput using low-NA focusing with a ∼5μm diameter focal spot. Such filters have low T<5% transmittance over a wide range of wavelengths 10 μm–0.5 mm (IR-THz). The high laser intensity ∼1 PW/cm${}^{2}$/pulse used for ablation cutting of THz filters reduces fabrication time, which can become an important factor due to hard X-ray radiation emitted from the plasma region [23]. With the increase in average power of ultra-short lasers equivalent to Moore’s law [27], direct laser writing is becoming not only a prototyping tool but a high throughput industrial fabrication technique [28].

We show that Kapton films of tens-of-μm thickness perform as FP etalons. By using different film thicknesses and controlling tilt with respect to the normal incidence, the possibility to tune wavelength over absorption bands of samples with high versatility can be realised. Reflection and transmission spectra from the FP etalon can both be used for spectral characterisation using the N${}^{2}$ITRO type setup as proposed in this study (Figure 5).

The polarisation and intensity distribution of the synchrotron THz beam is complex. With a characterised lateral distribution of polarisation components it would be possible to define a diffractive optical element for uniform mixing of polarisation and intensity over the focus, similar to what has been demonstrated for the optical spectral range [29]. Another solution for a more uniform distribution of polarisation over the focal region could be realised using a circular polariser based on total internal reflection in Fresnel rhomb. The average refractive index of Teflon directly measured by THz time domain spectroscopy (TDS) is n=1.48 and can be easily integrated into THz beamline at AuSy.

## Figures and Tables

**Figure 1 micromachines-13-01170-f001:**
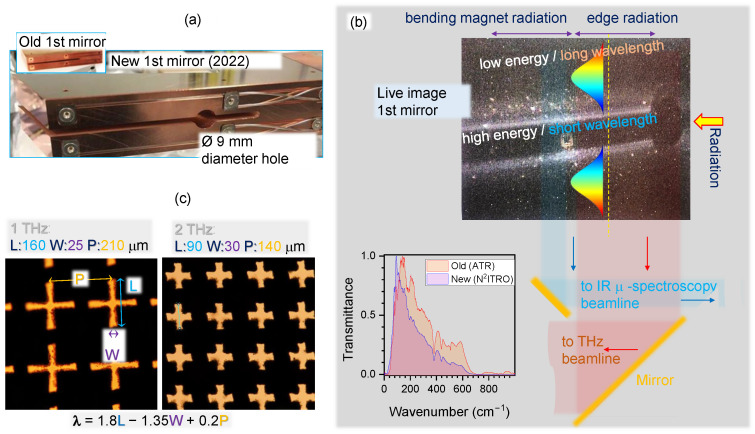
(**a**) Photo of the newly installed first mirror used for extracting the IR-THz radiation out of the storage ring at the AuSy (in use since January 2022). (**b**, **top**) Video camera feed of the 1st mirror with schematics of radiation split between two beamlines; visible light (part of synchrotron radiation) is apparent at the edges of the mirror. (**b**, **bottom**) Comparison of normalised transmittance observed before and after the replacement of the 1st mirror in ATR and Near-Normal Incidence Transmission and Reflection Optics (N${}^{2}$ITRO) modes, using a 6 μm Mylar beamsplitter (see Appendix A, Figure A1). (**c**) Optical micro-images of 1 and 2 THz filters laser ablated out of 20-μm-thick stainless steel SUS304 (L,W,P are the length, width, and period of the cross pattern).

**Figure 2 micromachines-13-01170-f002:**
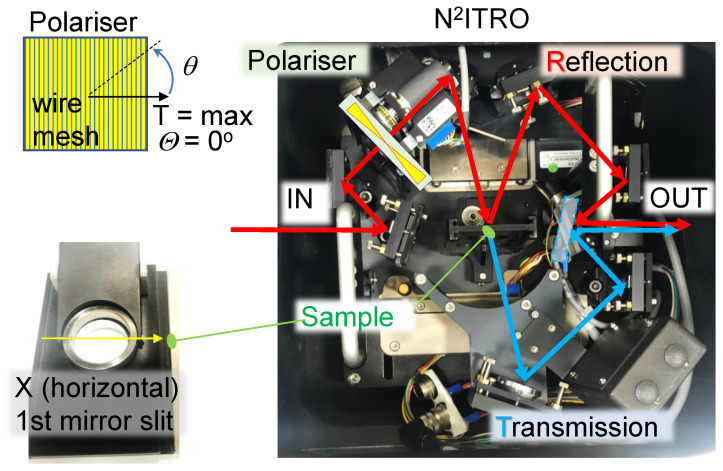
Beam tracing in the Near-Normal Incidence Transmission and Reflection Optics (N${}^{2}$ITRO, Bruker) unit with a step-motor controlled linear polariser. Measurements were carried out in vacuum ∼$10^{-5}$ atm with the fabricated filters placed at the sample holder. Polariser position for maximum transmittance *T* was set as $\theta =0^{\circ }$.

**Figure 3 micromachines-13-01170-f003:**
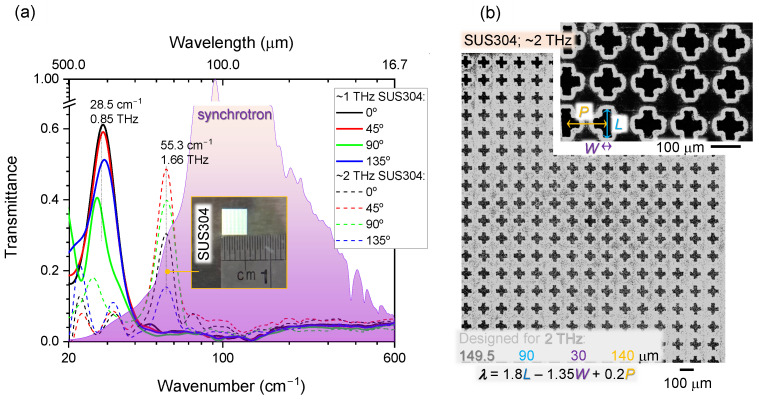
THz filters made out of 25-μm-thick stainless steel SUS304. (**a**) Transmittance spectra at four polarisation angles for ∼1 and ∼2 THz filters. The background shows normalised AuSy spectra (with the Mylar 6 μm beamsplitter used for detector). (**b**) SEM images of ∼2 THz filter.

**Figure 4 micromachines-13-01170-f004:**
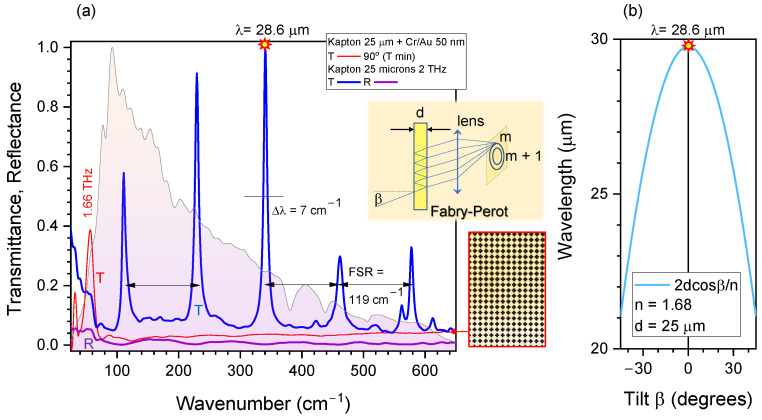
THz transmission and reflection studies using Kapton filters. (**a**) Transmission spectra show that metal-coated Kapton was performing equivalently to the SUS304 foil filters. Kapton with cut out crossed but without coating showed Fabry–Pérot (FP) etalon action with free spectral range FSR=119 cm${}^{-1}$; this was also observed with unstructured Kapton films. The background spectra is synchrotron radiation (detected without sample, filter nor polariser). Optical image of the metal-coated filter is shown on the right-side. The inset shows schematics of the FP etalon action. (**b**) Tunability of FP etalon by tilt angle β for the d=25μm thick Kapton film with n=1.68 refractive index. The condition of FP maxima is 2dcosβ=mλ.

**Figure 5 micromachines-13-01170-f005:**
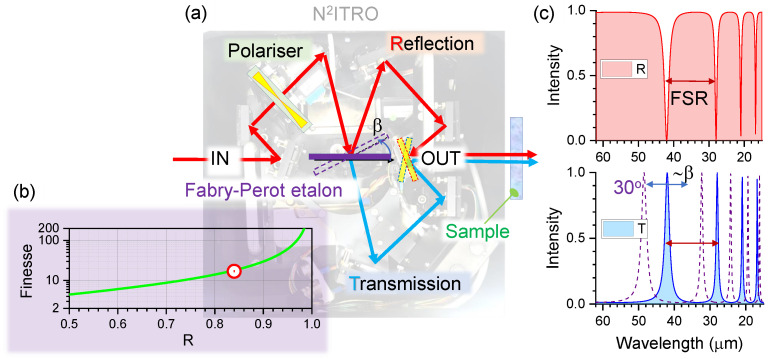
(**a**) Concept of T&R spectral analysis at broad and narrow spectral ranges using angular tunable FP etalon (thickness =d/cosβ) at tilt angle β using N${}^{2}$ITRO. (**b**) Finesse expression for $F_{c}>0.5$ is defined as $F_{c}\approx \pi \sqrt{R}/\left(1-R\right)$ where *R* is the reflectivity of Kapton/vacuum surface. In experiments, the finesse was $F_{c}\approx 17$, i.e., R=84% (a dot-marker). (**c**) The T,R spectra calculated for d=25μm and refractive index n=1.68 as in experiments with Kapton FP etalon; reflectivity was taken as R=0.8. Transmittance spectra at tilt angle of FP etalon $\cos\beta \equiv \cos30^{\circ }$ is shown. Note, FSR is not equally spaced in wavelength and equally spaced in energy (or wavenumber $\tilde{\nu }$) since λ∝1/ν.

**Figure 6 micromachines-13-01170-f006:**
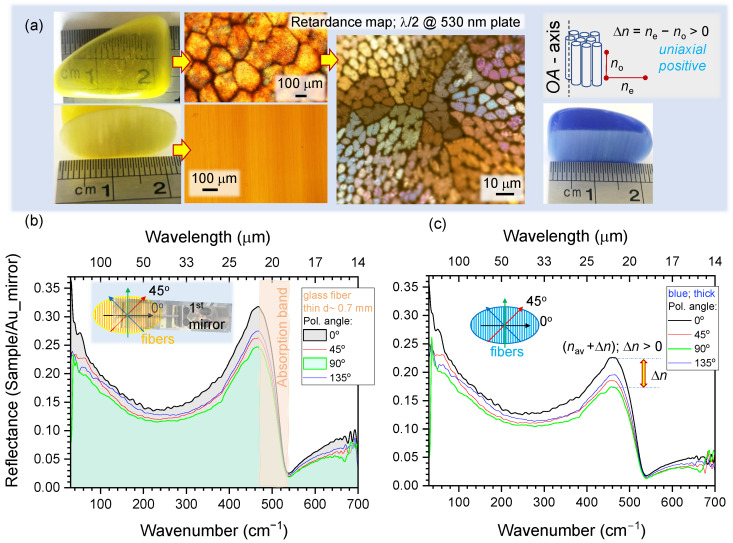
(**a**) “Cat-eye” silica fiber stone samples. Microscopy images were taken using cross Nicol polarisers with a λ/2 plate oriented at π/4 to the crossed polarisers; plate was for 530 nm wavelength to color render the retardance. Reflectance spectra measured with N${}^{2}$ITRO from silica-fiber “cat-eye” stones: yellow (**b**) and blue (**c**), which have form birefringence corresponding to the positive birefringence $\Delta n\equiv n_{e}-n_{o}>0$ classified as the uniaxial postive crystal. Input polarisation was set linear at four angles 0,π/4,π/2,and3π/4. Inset in (**b**) shows orientation of polarisation in respect to the slit in the first mirror. Anomalous dispersion at the absorption band is clearly discerned.

## Data Availability

The data presented in this study are available on request from the corresponding author.

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
