# Peer review of "THz Filters Made by Laser Ablation of Stainless Steel and Kapton Film"

_micromachines, 2022, doi:10.3390/mi13081170_

Round 1

Reviewer 1 Report

  This paper demonstrated a band-pass filter fabricated by femtosecond laser ablation with micro-foils of stainless steel and Kapton. Their spectral performance were tested in transmission and reflection at the THz beamline at the Australian Synchrotron. The proposed device maybe is useful for THz polarization-sensitive measurement, so I would recommend acceptance with minor revisions:

1. The more description about tilt compensation method in the machining process are needed to be mentioned, such as whether it is carried out on a three-axis platform, and whether the adjustment of focus is real-time or preset before machining.

2. Whether the thickness of Kapton FP etalon also affects the variation of the transmission spectrum. There should be a brief explanation of why 25 μm thick Kapton was chosen.

3. The measured details about how to change the polarization angle are not sufficient. The authors should provide more information.

Author Response

  1. The more description about tilt compensation method in the machining process are needed to be mentioned, such as whether it is carried out on a three-axis platform, and whether the adjustment of focus is real-time or preset before machining.

Expanded in Samples and Methods section A

  1. Whether the thickness of Kapton FP etalon also affects the variation of the transmission spectrum. There should be a brief explanation of why 25 μm thick Kapton was chosen.

Explanation added in Results and Discussion section B

  1. The measured details about how to change the polarization angle are not sufficient. The authors should provide more information.

Section about polarization added to the appendix. Mentioned in Samples and Methods section A

Reviewer 2 Report

In the manuscript, the cross-shaped apertures play a key role for the success of the proposed bandpass filters. As we know, the cross-shaped structures by laser ablation have been intensively investigated in the terahertz bandpass filters and optics for at least a decade. It is recommended that the authors make a comparison between their work and previous findings of the relevant studies listed below.

[1] B. Voisiat, A. Bičiūnas, I. et al., “Band-pass filters for THz spectral range fabricated by laser ablation,” Applied Physics A (2011).

[2] Hoang Le, Chandrasekhar Pradhani, et al., “Laser precession machining of cross-shaped terahertz bandpass filters,” Optics and Lasers in Engineering (2022).

[3] Rusnė Ivaškevičiūtė-Povilauskienė, L. Minkevičius, et al., “Flexible materials for terahertz optics: advantages of graphite-based structures,” Optical Materials Express

(2019).

[4] L. Minkevičius, K. Madeikis, B. et al., “Focusing Performance of Terahertz Zone Plates with Integrated Cross-shape Apertures,” (2014).

[5] V. Komarov and V. P. Meschanov, “Transmission properties of metal mesh filters at 90 GHz,” Journal of Computational Electronics (2019).

Author Response

Comparison added to Introduction with focus on femtosecond laser used in this work and the high power density.